# Learn Interpretable Word Embeddings Efficiently with von Mises-Fisher Distribution

## Abstract

Word embedding plays a key role in various tasks of natural language processing. However, the dominant word embedding model don't explain what information is carried with the resulting embeddings. To generate interpretable word embeddings we intend to replace the word vector with a probability density distribution. The insight here is that if we regularize the mixture distribution of all words to be uniform, then we can prove that the inner product between word embeddings represent the point-wise mutual information between words. Moreover, our model can also handle polysemy. Each word's probability density distribution will generate different vectors for its various meanings. We have evaluated our model in several word similarity tasks. Results show that our model can outperform the dominant models consistently in these tasks.

## 1 Introduction

Word embedding is a widespread technique in boosting the performance of modern NLP systems by learning a vector for each word as its semantic feature. The general idea of word embedding is to assign each word with a dense vector having lower dimensionality than the vocabularies' cardinality. In a qualified word embedding model, the vector-similarity tends to reflect the word-similarity. Therefore, feeding these vectors as features of words into the other NLP systems will always boost the performance of them in many downstream tasks (Turian et al., 2010; Socher et al., 2013).

One such qualified model is the skip-gram with negative sampling (SGNS) model proposed in word2vec (Mikolov et al., 2013; Joulin et al., 2016), which is very popular in various NLP tasks with expressive performance. The SGNS model propose to represent each word with vectors and estimate these vectors by applying maximum likelyhood estimation method. It implicitly factorize a word-context matrix containing a co-occurrence statistic. This would assign each word $w_c$ in the vocabulary with a "word" vector $\mathbf{v}_c \in \mathbb{R}^d$ and a "context" vector $\mathbf{u}_c \in \mathbb{R}^d$, so as to model the conditional probabilities $p(w_k|w_c)$ separately for different words. By maximizing the log likelyhood of $p(w_k|w_c)$, the SGNS model can estimate the "word" vector and "context" vector of each word.

However, the SGNS model's main problem is that it doesn't build interpretable model for the embeddings themself, and therefore, people don't understand how word vectors can express useful information. For example, previous work emphasized that the inner product between one word's "word" vector and another word's "context" vector represents the point-wise mutual information between the two words (Levy & Goldberg, 2014). However, people never use the "word-context" inner product in practise. Instead, people will simply chose "word" vectors as the word embeddings and drop the other one or vice versa. Therefore, we should care about the behaviour of "word-word" or "context-context" inner product, which are rarely analyzed and are never guaranteed to have any good properties.

In this paper, we propose a variational inference based framework to learn more interpretable word embeddings. That is, we would estimate a probability density distribution for each word instead of estimating a vector from the training corpus. To be specific, we propose to replace the "word" vector with a probability density distribution, namely the von Mises-Fisher (vMF) distribution, and keep the "context" vector for each word. As what we will show in this paper, representing the word with a probability density distribution can result in more interpretable word embeddings. Besides, the probability density representations can provide other benefits too. For example, such representations can model the polysemy phenomenon when we train our model. To be specific, we can sample a

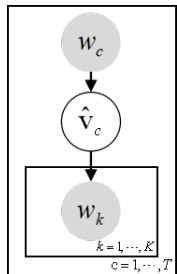

Figure 1: Each word $w_c$ would generate a latent meaning vector $\hat{\mathbf{v}}_c$, and then $w_k$ is picked by $\hat{\mathbf{v}}_c$.

vector from the vMF distribution to represent the specific meaning of this word in a particular context during training.

The main inspiration for this work is the analysis in Ma (2017)'s work, which can also be found in Arora et al. (2016)'s paper. They assumed that the word vectors would obey the uniform distribution over a sphere during the analysis. This assumption is critical in their analysis and yet it turns out to be wrong as reported by Mimno & Thompson (2017). In fact, the word frequency obey the Zipf's law, which means that it's impossible for word vectors to obey the uniform distribution when we represent each word by a vector. However, when we represent each word by a probability density distribution, it's possible that the mixture distribution of all words is uniform if we adjust each word's probability density distribution's position and shape carefully.

To estimate each word's probability density representation, we need to adopt the Bayesian variational inference technique, and this would result in a Bayesian version of SGNS model. We are not the first to propose a Bayesian version of word embedding model (Zhang et al., 2014; Sakaya et al., 2015; Barkan, 2017). Among them, the state-of-the-art model is the BSG model introduced by Bražinskas et al. (2017), and it is also the most related work with us. However, our model is different from the BSG model in many aspects. The BSG model represents each word with a Gaussian distribution while we adopt the vMF distribution. It's important to notice that the vMF distribution is defined over a unit hyper-sphere, which means that we will sample a unit vector for each word during training. Further more, the BSG model is based on the variational autoencoding framework (VAE). As reported by Davidson et al. (2018), it's impossible for the native VAE to work well when the prior is defined over a hyper-sphere. In contrast, we didn't adopt the VAE framework, and yet we would still talk about the reparameterization tricks. At last but not least, the BSG model focuses on how to build a more reasonable model using the VAE techniques, while we focus on how to generate more interpretable word embeddings using the vMF distribution.

Our main contributions can be summarized as follows. First, we proposed to represent each word with a vMF distribution. Second, we generated highly interpretable word embeddings, and show that our model's "context-context" inner-product would represent the point-wise mutual information between words. At last, our word embeddings out-perform the dominant models and the state-of-the-art model in various tasks.

## 2 OUR FRAMEWORK

In this section, we intend to generate interpretable word embeddings by adopting the vMF representation for each word. Specifically, we will show that the "context-context" inner-product between word embeddings would represent the point-wise mutual information between words.

### 2.1 MODEL DEFINITION

The SGNS model intends to maximize the probability of a context word appearing around a center word. It assumes this probability is proportional to the inner product between two fixed vectors. In contrast, we assume that the probability of a context word appearing around a center word should be the average probability when the center word take different meanings. That is, we are suggesting a generative model as pictured in 1. Assuming there are $T$ words $w_c$ in a training corpus, then for

each word we would like to predict the possible words $w_k$ appearing around it within a $K$ length word window. However, it's hard to determine what exactly the current word $w_c$ means, therefore, we propose to take the average probability over all possible $\hat{\mathbf{v}}_c$ for each observed $w_k$.

The training objective of our model is to find vector representations and probability density representations for words that are useful for predicting the surrounding words in a sentence. Given a sequence of training words $w_1, \cdots, w_T$, we are meant to maximize the average log likelyhood

$$\arg\max_{\theta} \frac{1}{T} \sum_{t=1}^{T} \sum_{j=-K/2}^{K/2} \log \int p(w_{t+j}|\hat{\mathbf{v}}_t, w_t; \theta) p(\hat{\mathbf{v}}_t|w_t; \theta) d\hat{\mathbf{v}} + L(\theta), \; j \neq 0, \tag{1}$$

where $\theta$ is the set of all parameters to be optimized. What's more, $\hat{\mathbf{v}}_t \in \mathbb{R}^d$ denotes a vector representation for a patential meaning of $w_t$, and $p(\hat{\mathbf{v}}_t|w_t; \theta)$ is the probability density representaton for $w_t$. We sample $\hat{\mathbf{v}}_t$ according to $p(\hat{\mathbf{v}}_t|w_t; \theta)$. At last, $p(w_{t+j}|\hat{\mathbf{v}}_t, w_t; \theta)$ denotes the probability of word $w_{t+j}$ appearing given $\hat{\mathbf{v}}_t$, and $L(\theta)$ denotes the possible regularization term.

Unfortunately, it's intractable to calculate the integration in (1). Therefore, we propose to git rid of the integration by applying the jensen inequality considering that $\log$ is concate and we are doing maximization. By doing so, our model can be defined as

$$\arg\max_{\theta} \frac{1}{T} \sum_{t=1}^{T} \sum_{j=-n}^{n} \mathbb{E}_{\hat{\mathbf{v}}_t \sim p(\hat{\mathbf{v}}_t^m|w_t)} \log p(w_{t+j}|\hat{\mathbf{v}}_t^m, w_t; \theta) + L(\theta), j \neq 0. \tag{2}$$

More details can be found in the appendix. We will call the $p(\hat{\mathbf{v}}_t^m|w_t)$ as a prior for each word $w_t$. Although the equation (2) is very similar to the ELBo in the standard varitional inference auto-encoder (VAE) technique, the actual meaning of it is quite different from ELBo. First, the expectation term in (2) doesn't involve a "encoder" as what VAE would do. Second, the regularization term in (2) is also different from the VAE's KL-divergence term, as what we will show.

## 2.2 EXPECTATION TERM

The most troublesome component of equation (2) is the epctation term. For each $\hat{\mathbf{v}}_t$ sampled from word $w_t$'s prior, we choose the softmax function to calculate the odds of word $w_{t+j}$ appearing around it. That is, we can decompose the expectation term into

$$\mathbb{E}_{\hat{\mathbf{v}}_t \sim p(\hat{\mathbf{v}}_t^m|w_t)} \log p(w_{t+j}|\hat{\mathbf{v}}_t^m, w_t; \theta) = \mathbb{E}_{\hat{\mathbf{v}}_t \sim p(\hat{\mathbf{v}}_t^m|w_t)} \left[ \log \frac{\exp(\mathbf{u}_{t+j}^\top \hat{\mathbf{v}}_t)}{\sum_{i=1}^{|V|} \exp(\mathbf{u}_i^\top \hat{\mathbf{v}}_t)} \right], \tag{3}$$

where $\mathbf{u}_i \in \mathbb{R}^d$ denotes the "context" vector for word $w_i$, and $|V|$ is the cardinality of our vocabulary. As what's been suggested in the SGNS model, we extend the negative sampling technique to our model to avoid the computation of the softmax function's denominator. It's worth to notice that different words would have different denominators in theory. We will use $Z_t$ to denote the denominator of word $w_t$. According to the theory proved by Gutmann & Hyvärinen (2012), maximizing objective (3) is equivalent to minimizing

$$\mathbb{E}_{\hat{\mathbf{v}}_t \sim p(\hat{\mathbf{v}}_t^m|w_t)} \left[ \log \sigma(\mathbf{u}_{t+j}^\top \hat{\mathbf{v}}_t) + N\mathbb{E}_{i \sim p(w_i)} \log \sigma(-\mathbf{u}_i^\top \hat{\mathbf{v}}_t) \right], \tag{4}$$

where $\mathbf{u}_i$ is the context vector of word $w_i$, and $w_i$ is the negative word sampled according to the empirical unigram probability $p(w_i)$. There are $N$ negative samples.

At last, we also need to sample $\hat{\mathbf{v}}_t$, and we choose von Mises-Fisher (vMF) distribution to calculate the probability of $w_t$ taking this particular meaning. The vMF distribution is an analogy of Gaussian distribution over the unit sphere. It's parameterized by a mean vector and a concentration parameter $\kappa \geq 0$. For word $w_t$, it's corresponding mean vector $\mathbf{v}_t$ can be interpreted as the vector representation for its average meaning. Formally, the probability of $w_t$ taking one particular meaning is

$$\begin{aligned} p(\hat{\mathbf{v}}_t|w_t) &= c_d(\kappa_t) \exp(\kappa_t \mathbf{v}_t^\top \hat{\mathbf{v}}_t), \\ c_d(\kappa_t) &= \frac{\kappa_t^{d/2-1}}{(2\pi)^{d/2} I_{d/2-1}(\kappa_t)}, \end{aligned} \tag{5}$$

where $d$ is the dimension of embedding space and $I_{d/2-1}(\cdot)$ denotes the modified Bessel function of the first kind Sra (2016).

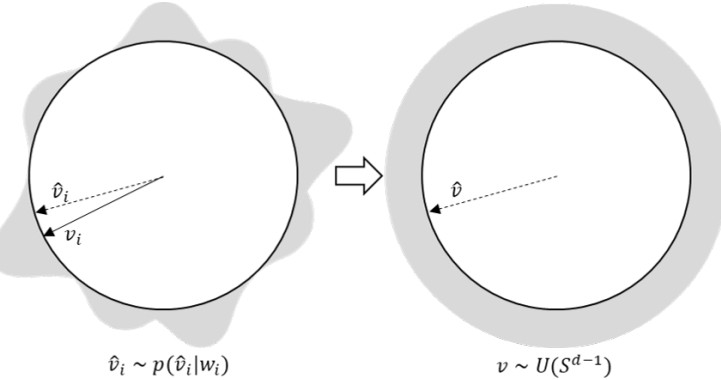

$$\hat{v}_i \sim p(\hat{v}_i|w_i) \qquad\qquad v \sim U(S^{d-1})$$

Figure 2: By adjusting the position and shape of each word's vMF distribution, the mixture distribution of all words can be uniform.

## 2.3 REGULARIZATION TERM

We will introduce an regularization term in this subsection, and show that this term will link the "context-context" inner-product directly to the point-wise mutual information. The general idea is to regularize the mixture distribution of all words to be uniform, then we can achieve our goal. This can be done by simply applying the maximum entropy algorithm which would result in a uniform distribution naturally. Therefore, the regularization term we are seeking for is

$$
\begin{aligned}
\arg\max_{\theta} \quad & p(\hat{\mathbf{v}})\log p(\hat{\mathbf{v}}), \\
where \quad & p(\hat{\mathbf{v}}) = \sum_{i=1}^{|V|} p(w_i)p(\hat{\mathbf{v}}|w_i).
\end{aligned}
\tag{6}
$$

Objective (4) + (6) is the final objective that we want to optimize. We claim that by doing so, for each pair of words $w_c$ and $w_k$, we have

$$
\frac{\mathbf{u}_k^\top \mathbf{u}_c}{d} \approx PMI(w_k, w_c) := \log\left(\frac{p(w_k, w_c)}{p(w_k)p(w_c)}\right),
$$

where $PMI(w_k, w_c)$ is called the point-wise mutual information (PMI) between word $w_k$ and word $w_c$, and $p(w_k, w_c)$ denotes the probability for them to appear in the same word window.

Before we show why our claim holds, we would like to emphasize why PMI is important. PMI indicates how much more possible that word $w_k, w_c$ co-occur than by chance (Church & Hanks, 1990). Then, people developed the notion of word window to help define when two words "co-occur", i.e., $w_k$ and $w_c$ co-occur, only when they appear in the same word window. Most of the word embedding model will take advantage of such co-occurrence statistics. Indeed, the PMI evaluated from co-occurrence counts has a strong linear relationship with human semantic similarity judgments from survey data (Hashimoto et al., 2016). In conclusion, it's reasonable to relate word embedding with the point-wise mutual information.

Then, we will show how to link the "context-context" inner-product directly to the PMI by two steps. The first step is to show that

$$
\begin{aligned}
\log p(w_k, w_c) \quad &\approx \frac{\|u_k + u_c\|_2^2}{2d} - 2\log(Z), \\
\log p(w_k) \quad &\approx \frac{\|u_k\|_2^2}{2d} - \log(Z), \\
\log p(w_c) \quad &\approx \frac{\|u_c\|_2^2}{2d} - \log(Z),
\end{aligned}
\tag{7}
$$

where $Z$ is the constant that most $Z_c$ approximate to. This is a conclusion proved by Ma (2017). The second step is quite obvious because

$$
PMI(w_k, w_c) = \log p(w_k, w_c) - \log p(w_k) - \log p(w_c) \approx \frac{\mathbf{u}_k^\top \mathbf{u}_c}{d}.
$$

We will show (7) by a series equations briefly, and reveal why the regularization term is very important. More details can be found in the appendix. We start with $p(w_k, w_c)$

$$
\begin{aligned}
p(w_k, w_c) &= \sum_{i=1}^{|V|} p(w_i)p(w_k, w_c|w_i) \\
&= \sum_{i=1}^{|V|} p(w_i)p(w_k|w_i)p(w_c|w_i) \\
&= \mathop{\mathbb{E}}_{i \sim p(w_i)} p(w_k|w_i)p(w_c|w_i) \\
&= \mathop{\mathbb{E}}_{i \sim p(w_i)} \int \frac{e^{\mathbf{u}_k^\top \hat{\mathbf{v}}_i'}}{Z_i} p(\hat{\mathbf{v}}_i'|w_i)ds' \int \frac{e^{\mathbf{u}_c^\top \hat{\mathbf{v}}_i}}{Z_i} p(\hat{\mathbf{v}}_i|w_i)ds \\
&\approx \frac{1}{Z^2} \mathop{\mathbb{E}}_{i \sim p(w_i)} \int \int e^{\mathbf{u}_k^\top \hat{\mathbf{v}}_i'} e^{\mathbf{u}_c^\top \hat{\mathbf{v}}_i} p(\hat{\mathbf{v}}_i'|w_i)p(\hat{\mathbf{v}}_i|w_i)dsds' \\
&\approx \frac{1}{Z^2} \mathop{\mathbb{E}}_{i \sim p(w_i)} \int \exp\left[(\mathbf{u}_k + \mathbf{u}_c)^\top \hat{\mathbf{v}}_i\right] p(\hat{\mathbf{v}}_i|w_i)ds \\
&= \frac{1}{Z^2} \sum_{i=1}^{|V|} p(w_i) * \left\{ \int \exp\left[(\mathbf{u}_k + \mathbf{u}_c)^\top \hat{\mathbf{v}}_i\right] p(\hat{\mathbf{v}}_i|w_i)ds \right\} \\
&= \frac{1}{Z^2} \int \exp\left[(\mathbf{u}_k + \mathbf{u}_c)^\top \hat{\mathbf{v}}\right] \left[ \sum_{i=1}^{|V|} p(w_i)p(\hat{\mathbf{v}}|w_i) \right] ds \\
&= \frac{1}{Z^2} \int \exp\left[(\mathbf{u}_k + \mathbf{u}_c)^\top \hat{\mathbf{v}}\right] p(\hat{\mathbf{v}})ds \\
&= \frac{1}{Z^2} \mathop{\mathbb{E}}_{\hat{\mathbf{v}} \sim p(\hat{\mathbf{v}})} \left\{ \exp\left[(\mathbf{u}_k + \mathbf{u}_c)^\top \hat{\mathbf{v}}\right] \right\} \\
&= \frac{1}{Z^2} \mathbb{E}_{x \sim \mathcal{N}(0, \|\mathbf{u}_k + \mathbf{u}_c\|_2^2/d)} \{\exp(x)\} \\
&\approx \frac{1}{Z^2} \exp\left( \frac{\|\mathbf{u}_k + \mathbf{u}_c\|_2^2}{2d} \right).
\end{aligned}
\tag{8}
$$

Step ten is why we need a regularization term, but before that, we would like to explain all the equations above. The first step of (8) says that $w_k$ and $w_c$ co-occur iff they appear in another word $w_i$'s word window together. This is true for the Skip-gram model. The second step is also true when we considering the definition of $p(w_k|w_i)$ and $p(w_c|w_i)$ in the Skip-gram model. Step four is just by definition and we use slightly different notations here to indicate that $\hat{\mathbf{v}}_i, \hat{\mathbf{v}}_i'$ are different variable. Step six is a strong claim which needs rigorous prove. We put this prove in the appendix. The key insight is that $\hat{\mathbf{v}}_i, \hat{\mathbf{v}}_i'$ obey the same vMF distribution, which means that when this vMF is concentrate enough, then the probability of $\hat{\mathbf{v}}_i, \hat{\mathbf{v}}_i'$ being very different is small. In the eighth step, we omit $i$ to emphasize that every word can generate the same vector $\hat{\mathbf{v}}$ and that's why the summation and integration can exchange with each other in this way.

If we regularize the $\hat{\mathbf{v}}$ to be uniformly distributed over the unit sphere in step ten, then $(\mathbf{u}_k + \mathbf{u}_c)^\top \mathbf{v}$ will obey Gaussian distribution approximately (Ma, 2017). This means that step eleven holds. More details can be found in the appendix. The last step is the result of a famous calculation practise

$$
\mathbb{E}_{x \sim \mathcal{N}(0, \sigma^2)}\{\exp(x)\} = \exp(\sigma^2/2).
$$

Obviously, by replacing the "word" vector with the vMF distribution, we can eliminate the assumption that $p(w_i)$ being uniform.

## 2.4 REPARAMETERIZATION TRICK

There is one problem to solve before our model becomes practical. $p(\hat{\mathbf{v}}_t|w_t)$ is difficult to optimize because the operation of sampling is nondifferentiable. We can solve this problem by applying the reparameterization trick. The vMF distribution's reparameterization trick is usually discussed in the context of hyperspherical variational auto-encoders (Davidson et al., 2018; Xu & Durrett, 2018; Guu et al., 2018). To simplify our model, we propose to fix the concentration parameter $\kappa_t$ as a constant during training for each word $w_t$. This is because the gradient estimation of $\kappa$ is complex and computational expensive.

When it comes to each word $w_t$'s mean vector $\mathbf{v}_t$, we follow the technique used in Xu & Durrett's work. Firstly, we sample an auxiliary random variable $\omega$ according to the rejection sampling scheme of Wood (1994). The distribution of $\omega$ is controlled by $\kappa$. Specifically, the probability of $\omega$ being sampled is

$$
p(\omega; \kappa) \propto \exp(\omega\kappa)(1 - \omega^2).
$$

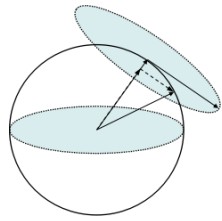

Figure 3: The illustration about our reparameterization trick.

Then we draw our $\hat{\mathbf{v}}_t$ in the following way

$$\hat{\mathbf{v}}_t = \omega \mathbf{v}_t + \mathbf{z}\sqrt{1 - \omega^2},$$

where $\mathbf{z}$ is a random unit vector and is tangent to the unit sphere $\mathbb{S}^{d-1}$ at $\mathbf{v}_t$. Figure 3 illustrates the geometric vision. Because of applying this trick, we can take gradient with respect to $\mathbf{v}_t$ as usual.

### 2.5 FINAL ALGORITHM

Putting all the details together, we have the following objective to minimize

$$\underbrace{\log \sigma(\mathbf{u}_k^\top \hat{\mathbf{v}}) + N \mathbb{E}_{i \sim p(w_i)} \left[ \log \sigma(-\mathbf{u}_i^\top \hat{\mathbf{v}}) \right]}_{L_1} - \underbrace{p(\hat{\mathbf{v}}) \log p(\hat{\mathbf{v}})}_{L_2},$$

$$
\begin{aligned}
where \quad & \hat{\mathbf{v}} = \omega \mathbf{v}_c + \mathbf{z}\sqrt{1 - \omega^2}, \; \|\mathbf{v}_c\|_2 = 1 \\
& \omega \sim p(\omega; \kappa_c) \propto \exp(\omega \kappa_c)(1 - \omega^2), \\
& \|\mathbf{z}\|_2 = 1, \mathbf{z} \; is \; tangent \; to \; \mathbb{S}^{d-1} \; at \; \mathbf{v}_c, \\
& p(\hat{\mathbf{v}}) = \sum_{j=1}^{|V|} p(w_j) p(\hat{\mathbf{v}}|w_j) \\
& p(\hat{\mathbf{v}}|w_j) = c_d(\kappa_j) \exp(\kappa_j \mathbf{v}_j^\top \hat{\mathbf{v}}), \\
& c_d(\kappa_j) = \frac{\kappa_j^{d/2-1}}{(2\pi)^{d/2} I_{d/2-1}(\kappa_j)}.
\end{aligned}
\tag{9}
$$

Based on our final model, we propose algorithm (1) to train the word embeddings. The **for** loop started from line 1 controls how many times we will go through the entire training corpus. Then, for each epoch, we would iterate over each word in the corpus as what the **for** loop in line 2 suggests. In line 3, we will take $w_c$'s mean vector $\mathbf{v}_c$ and concentrate parameter $\kappa_c$ from dictionaries. The third layer of **for** loop will iterate over all the words in a word window centered around $w_c$. For each word $w_k$ in this word window, we will take its context vector $\mathbf{u}_k$ from the dictionary, and sample a $\hat{\mathbf{v}}$ from $w_c$'s prior with the help of our reparameterization trick (line 5 to line 6). Then form line 7 to line 10, we will sample $N$ negative samples for each $w_k$, and then calculate the gradients of $L_1$. From line 11 to line 13, we will calculate $L_2$ based on the $\hat{\mathbf{v}}$, and update all word's prior accordingly.

---

**noend 1** PDF $(N, \mathbf{u}_i, \mathbf{v}_i, \kappa_i, p(w_i), i = 1, \cdots, |V|)$

---

1: **for** $epoch$ in $epochs$ **do**
2:    **for** $w_c$ in $corpus$ **do**
3:      $\mathbf{v}_c \leftarrow \{\mathbf{v}_i\}_{i=1,\cdots,|V|}, \kappa_c \leftarrow \{\kappa_i\}_{i=1,\cdots,|V|}$
4:      **for** $w_k$ in $Window(w_c)$ **do**
5:        $\omega \sim p(\omega; \kappa_c), \mathbf{z} \sim \mathbf{U}(\mathbb{S}^{d-2})$
6:        $\hat{\mathbf{v}} \leftarrow \omega \mathbf{v}_c + \mathbf{z}\sqrt{1 - \omega^2}, \mathbf{u}_k \leftarrow \{\mathbf{u}_i\}_{i=1,\cdots,|V|}$
7:        **for** $i$ in range($N$) **do**
8:          $w_i \sim p(w_i), \mathbf{u}_i \leftarrow \{\mathbf{u}_i\}_{i=1,\cdots,|V|}$
9:          $L_1 \leftarrow L_1(\mathbf{u}_k, \mathbf{u}_i, \hat{\mathbf{v}})$
10:        Update $\mathbf{v_c}, \mathbf{u}_k$ according to $L_1$'s gradient.
11:        $L_2 \leftarrow L_2(\{\mathbf{v}_j\}_{j=1,\cdots,|V|}, \hat{\mathbf{v}})$
12:        **for** $j$ in range($|V|$) **do**
13:          Update $\mathbf{v}_j$ according to $L_2$'s gradient.

---

| Model | Embedding | WS353 | WS353-SIM | WS353-REL | RG65 | MEN |
|-------|-----------|-------|-----------|-----------|------|-----|
| SGNS | context | 0.5648 | 0.6184 | 0.4832 | 0.491 | 0.479 |
| GloVe | context | 0.5686 | 0.6219 | 0.4864 | 0.4959 | 0.4921 |
| Ours | context | **0.6378** | **0.6932** | **0.5961** | **0.522** | **0.646** |
| SGNS | word | 0.6606 | 0.7071 | 0.6237 | **0.669** | 0.676 |
| GloVe | word | 0.6436 | 0.7085 | 0.5865 | 0.6606 | 0.666 |
| Ours | mean | **0.6637** | **0.7444** | **0.6404** | 0.596 | **0.68** |

Table 1: Results on the word similarity tasks.

## 3 EXPERIMENTS

We will now experimentally validate our embedding by comparing the performance of our model with the dominant word embedding models on the word similarity tasks. Subsection 3.1 presents all the experiment settings for different training corpus and different word embedding models. Subsection 3.2 introduces several word similarity benchmarks, and how we evaluate model's performance on them. Subsection 3.3 compares our model with the STOA model on several benchmarks.

### 3.1 EXPERIMET SETTINGS

We use part of massachusetts bay transportation authority (mbta) web crawled corpus because it is well cleaned and is large enough. The mbta corpus contains about 600 million tokens. We preprocessed all corpora by removing non-English characters, numbers and lower-casing all the text. The vocabulary was restricted to the 100K most frequent words in each corpus.

We trained embeddings using three methods: word2vec Mikolov et al. (2013), GloVe Pennington et al. (2014), and our model. This is because these models are implemented with C/C++, and therefore are fast enough to be evaluated on mbta corpus. For fairness we fix all hyperparameters for word2vec, GloVe and our model. Specifically, we trained for 5 epochs for each model using 75 threads for parallel computation; the word embedding dimension is 100; the window size is 5.

For the word2vec, and our model, the negative sampling number is 5; We adopt the Hogwild! algorithm to train models, and the learning rate decays linearly from 0.0025 to 0. The noise distribution is set as the same as used in Mikolov et al., $p_n(w) \propto p(w_w)^{0.75}$. We also use a rejection threshold of $10^{-4}$ to subsample the most frequent words.

For the GloVe model, we follow the original inplementation's default settings for the other hyperparameters. This means the initial learning rate is 0.05.

We also use the news corpus[1] with about 15 million tokens to evaluate the BSG model. This is because the original implementation of BSG is based on theano, and it's slow to be trained on the mbta corpus. Given the small volume of this corpus, we trained for 25 epochs for each model using 12 threads. we also restrict each model's vocabulary to be 10K. For both the BSG, word2vec and our model, we set the negative number to be 10; the window width is 10.

### 3.2 PERFORMANCE EVALUATION

We test the quality of the word embeddings by checking if our word embeddings agree with the human judgement on word similarity / relatedness.

For the mbta corpus, we test the performance of our embeddings on three benchmarks: the Word Similarity353 data set Finkelstein et al. (2002), the RG65 data set Luong et al. (2013), and the MEN data set Bruni et al. (2014). Taking the WS353 data set for example, it contains 353 word pairs along with their similarity scores assigned by 29 subjects. These subjects possessed near-native command of English and they are instructed to estimate the relatedness of the words in pairs on a scale from 0 to 10. During experiments, we will cumpute the Spearman's rank correlation coefficient Spearman (1904) between human judgement and the inner product between the vector representations. The larger this coefficient is, the better this embedding is and the mximume value is 1. Some words in

---

[1] https://drive.google.com/file/d/1QWC2x6qq8KyHFUCgyvVJJoGHexZrw7gO/view

| Datasets | Ours | BSG | GloVe | SGNS |
|---|---|---|---|---|
| MC-30 | **0.6190** | 0.5818 | 0.4524 | 0.4524 |
| MEN-Tr-3k | 0.4905 | 0.3937 | 0.3607 | **0.5182** |
| MTurk-287 | 0.5343 | 0.5147 | 0.3334 | **0.5351** |
| MTurk-771 | **0.4221** | 0.3693 | 0.2647 | 0.4177 |
| RG65 | 0.5727 | **0.6036** | 0.3455 | 0.4000 |
| RW-STNFRD | **0.4172** | 0.3830 | 0.2710 | 0.3725 |
| SIMLEX-999 | 0.2110 | 0.1717 | 0.1914 | **0.2339** |
| VERB-143 | **0.3127** | 0.1371 | 0.0712 | 0.2376 |
| WS353-ALL | **0.4567** | 0.4350 | 0.2379 | 0.4387 |
| WS353-Rel | 0.3485 | **0.4197** | 0.1583 | 0.3591 |
| WS353-SIM | **0.6070** | 0.5018 | 0.3277 | 0.5549 |

Table 2: Results on the word similarity tasks.

these testing datasets do not appear in our training corpora, and this means we can't calculate the inner product between vectors for those words. In order to provide comparable results, we propose to use the mean vectors of the rest words for these missing words. Also, the mean vector of each word's prior is similar to the "word" vector in the SGNS model, and we also test the performance of this vector too.

We also evaluate the performance over more Benchmarks. Table 2 presents similarity results computed using the online tool of Faruqui & Dyer (2014). Since the BSG model can only generate the mean vectors from its prior, we only evaluate the "word" or mean vectors in these experiments.

## 3.3 COMPARE WITH THE DOMINANT MODELS

Table 1 shows the results on these benchmarks. As we can see, our "context" word embedding can constantly outperform the counterpart of the dominant word embedding models by a large margin. This is exactly what we expect according to our theory. It's interesting that the "word" vector of our model can still outperform ours "context" vector. Another exception is that on the RG65 test set, the SGNS model can outperform our model by a large margin. We argue this may be caused by its bias – it only contains 65 noun pairs after all. Besides, our model's "context" vector can still outperform SGNS on this test set.

Also, the gap between the "context" vectors and the "word" vectors in our model is much smaller than the dominant models' gaps. These results demonstrate that our model is more specific about which part of our embedding contains the useful information.

## 3.4 COMPARE WITH THE BSG MODEL

First, we observe that our model can out-perform the BSG model for almost every task except for the RG65 data set as before. Although the BSG model can perform better than our model in the WS353-Rel data set, we would like to point out that this data set is a subset of WS353-ALL. Therefore, this may also be caused by the bias of small data set. Second, the performance of BSG model is weaker than the SGNS model for some data sets. Given that we used their released implementation directly, this may be the result of small training data set, i.e., the SGNS model may perform better in the case of small training data set because of its simplicity.

## 4 CONCLUSIONS

We generated interpretable word embeddings by represent each word with a von Mises-Fisher distribution. We have demonstrated that our word embeddings can be linked to the point-wise mutual information directly without making any unrealistic assumptions. The experiments over different training and testing data sets demonstrate that our model can outperform both the dominant and the STOA models. We argue that our insight into the interpretable word embeddings is important. For example, as we are sure that the unit word vectors can encode the semantic similarity between words, it's possible to encode the syntactic information between words into the norm of word vectors.

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
