# OpenReview forum: "Learn Interpretable Word Embeddings Efficiently with von Mises-Fisher Distribution"
_ICLR.cc/2020/Conference — Reject_

### Official Review · AnonReviewer3 · 2019-10-20
**Official Blind Review #3**

**Rating:** 1

**Review:**

Summary: This paper proposed a variational word embedding method, by using vMF distribution as prior and adding an entropy regularization term.

Strengths:
[+] In the experiment parts, the authors describe the experiment settings in detail.
[+] Detailed descriptions for each equation is given.

Weaknesses:
[-] Template: It seems that the authors use the wrong template, which is not the template for ICLR2020.
[-] Appendix: Although the author mentioned 'appendix' many times, I cannot see the appendix.
[-] Motivation: The connection between motivation and proposed method seems weak. The authors argue 'interpretable' in their title and abstract, but their method and experiments do not show this point explicitly.

Questions
[.] The experiments seem a little weak. The vocabulary size is 10K and the corpus is not so big, and I wonder whether the performance of the proposed method will be better for large corpus and longer training time.
[.] For Equation 8, we should have a guarantee on the concentration of partition functions. Is it still true for vMF distribution?
[.] What is the advantage of vMF distribution?

**Experience Assessment:**

I have published one or two papers in this area.

**Review Assessment: Checking Correctness Of Derivations And Theory:**

I assessed the sensibility of the derivations and theory.

**Review Assessment: Checking Correctness Of Experiments:**

I assessed the sensibility of the experiments.

**Review Assessment: Thoroughness In Paper Reading:**

I read the paper thoroughly.

---

### Official Review · AnonReviewer1 · 2019-10-23
**Official Blind Review #1**

**Rating:** 8

**Review:**

The paper addresses the problem the problem in word embeddings where "word-word" or "context-context" inner products are relied upon in practice when the embeddings are usually only optimized for good properties of "word-context" inner products. The new approach to training word embeddings addresses interpretability in the sense of having a theoretically grounded interpretation for the good properties that the inner products should possess -- namely that they capture the PMI between words. By changing the embedding training procedure (to include a step that samples vectors from each word's distribution representation) this interpretation is made possible without problematic theoretical assumptions. Now trained for the purpose they are intended to be used for, the approach yields strong SOTA results on the non-small challenge datasets.

The word is well exceptionally well motivated (we should directly optimize for the properties we want!) and situated in the literature. The opposition of Arora 2016 (getting at "interpretable" word embeddings with a latent variable model) and Mimno & Thompson 2017 (problematic assumptions in skip-gram embeddings) is particularly convincing. The vMF distribution is underexamined as tool for analyzing high-dimensional semantic vector distributions, and it is an excellent fit for the purposes of this project.

To back up claims of theoretical interpretability, a derivation (building on Ma 2017) proceeds without problematic leap thanks to a property introduced by a carefully selected (if straightforward) regularization term. This reviewer did not read the appendix (not sure where this would be located -- first time reviewing with OpenReview here), but the intuition doesn't seem to be much different than Ma.

To back up claims of applied performance, appropriate experiments are conducted on multiple evaluation datasets. The results seem to be fairly interpreted.

This reviewer decides to accept this paper for its balance of theoretical and empirical contributions and for the role it might play in reducing dependence on mysticisms in word embeddings (relying on accidental / uncharacterized properties of previous embedding strategies).

Suggestions for improvement:
- Run a spell checker. It will catch a large number of problems that weren't ever big enough to hurt my appreciation of the paper.
- Consider the creation of an adversarial dataset (of ~3 words in ~3 contexts) where techniques that optimize for the wrong thing will catastrophically fail but the proposed approach succeeds.
- Write one or two more sentences about the fate of \kappa_c -- is it ever updated or am I just missing something? Making a bigger point about leaving \kappa un-optimized shows there is room for additional depth in this line of thinking. (If you start learning concentration parameters, check out the Kent distribution for an even more expressive distribution on the unit hypersphere: https://en.wikipedia.org/wiki/Kent_distribution)
- Figure out what happened to your appendix.
- Watch out that readers/reviewers of similar work may try to read the word "interpretable" in the title as a reference to the much broader topic of interpretability in ML related to explaining a model's behavior. Is there a synonym for "interpretable" that won't raise this false link?

**Experience Assessment:**

I have read many papers in this area.

**Review Assessment: Checking Correctness Of Derivations And Theory:**

I assessed the sensibility of the derivations and theory.

**Review Assessment: Checking Correctness Of Experiments:**

I assessed the sensibility of the experiments.

**Review Assessment: Thoroughness In Paper Reading:**

I read the paper at least twice and used my best judgement in assessing the paper.

---

### Official Review · AnonReviewer4 · 2019-10-31
**Official Blind Review #4**

**Rating:** 1

**Review:**

This paper is about learning word embeddings. In contrast to previous work (word2vec's Skipgram) where a word is represented by a fixed word and context vector, here each word is represented by a von Mises-Fisher distribution and a context vector.
The paper claims that the resulting embeddings are more interpretable, and that the inner product of two context vectors represents their point-wise mutual information.

My main concerns are the following:

- Representing words by distributions is not a novel idea; it was previously done by Brazinskas et al. (2017): Embedding words as distributions with a bayesian skip-gram model (BSG). They however used Gaussian distributions and not von Mises-Fisher. BSG is acknowledged in the paper, but only small scale comparisons are performed (15 million) while the BSG paper uses a lot larger data sets. There is therefore not a meaningful comparison to the most relevant previous work.
- Word similarity experiments are not enough to justify this approach. BSG at least showed the strength of using distributions to represent words by showing that different samples could constitute different word senses/meanings. There is no such analysis here.
- The claim that the resulting representations are more "interpretable" is not backed up by any evidence at all, even though the word "interpretable" is in the title and the list of contributions.

Generally this paper could benefit from proof reading and editing.

**Experience Assessment:**

I have read many papers in this area.

**Review Assessment: Checking Correctness Of Derivations And Theory:**

I did not assess the derivations or theory.

**Review Assessment: Checking Correctness Of Experiments:**

I assessed the sensibility of the experiments.

**Review Assessment: Thoroughness In Paper Reading:**

I read the paper at least twice and used my best judgement in assessing the paper.

---

### Decision · Program_Chairs · 2019-12-19

**Decision:**

Reject

**Comment:**

The paper presents an approach to learning interpretable word embeddings. The reviewers put this in the lower half of the submissions. One reason seems to be the size of the training corpora used in the experiments, as well as the limited number of experiments; another that the claim of interpretability seems over-stated. There's also a lack of comparison to related work. I also think it would be interesting to move beyond the standard benchmarks - and either use word embeddings downstream or learn word embeddings for multiple languages [you should do this, regardless] and use Procrustes analysis or the like to learn a mapping: A good embedding algorithm should induce more linearly alignable embedding spaces.

NB: While the authors cite other work by these authors, [0] seems relevant, too. Other related work: [1-4].

[0] https://www.aclweb.org/anthology/Q15-1016.pdf
[1] https://www.aclweb.org/anthology/Q16-1020.pdf
[2] https://www.aclweb.org/anthology/W19-4329.pdf
[3] https://www.aclweb.org/anthology/D17-1198/
[4] https://www.aclweb.org/anthology/D15-1183.pdf